# Unsupervised Machine Learning for Anomaly Detection in LHC Collider Searches

**Antonio D'Avanzo [1, 2], on behalf of the ATLAS Collaboration**

**1** University of Naples "Federico II", Naples, Italy
**2** Istituto Nazionale di Fisica Nucleare (INFN), Frascati, Italy

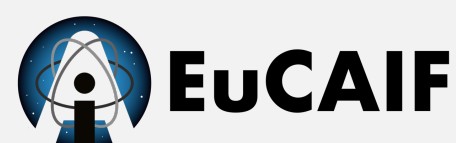

*The 2nd European AI for Fundamental Physics Conference (EuCAIFCon2025) Cagliari, Sardinia, 16-20 June 2025*

## Abstract

**Searches for new physics at the LHC at CERN traditionally use advanced simulations to model Standard Model and new-physics processes in high-energy collisions and compare them with data. The lack of recent direct discoveries, however, has motivated the development of model-independent approaches in HEP to complement existing hypothesis-driven analyses, particularly Anomaly Detection. A review of the latest efforts in BSM searches with anomaly detection is presented in these proceedings, focusing on contributions within the ATLAS collaboration at LHC and discussing Variational Recurrent Neural Network, Deep Transformer and Graph Anomaly Detection applications.**

## 1 Anomaly Detection at the LHC with the ATLAS experiment

Anomaly Detection (AD) [1] refers to the set of Machine Learning (ML) techniques used to spot outliers in a given collection of data. It's a widely spread approach in many scientific fields and topics, thanks to the potential generalization of a classification task reduced to the identification of many unknown classes against one well-known class. This includes HEP, due to the absence of breakthroughs in the searches for Beyond Standard Model (BSM) physics at the particle accelerator experiments worldwide, including the ATLAS experiment at the LHC at CERN [2]. ATLAS is a multi-purpose experiment designed to exploit the LHC full discovery potential due to the many Standard Model open questions, e.g. dark matter and gravitational interaction, which drive the efforts for BSM searches that are are carried out, in the majority of scenarios, in a model-dependent way. However, due to the scarce sensitivity to more than one process, complementary AD based model-independent searches have also been performed. This enabled the identification of anomalous events that deviate significantly from the expected background-only hypothesis, typically appearing as "bumps" in the final

fit physical observable, without assuming a specific new-physics model or prior theoretical expectations. In the next sections, the first ATLAS analysis performed in a model-independent way using a fully-unsupervised AD algorithm is discussed, followed by a dissertation of the subsequent *R&D* developed on open data for a similar and updated AD ATLAS search.

## 2  Unsupervised AD models in ATLAS: first steps with $Y \rightarrow XH$

A search for a heavy resonance $Y$ ($\approx$ TeV) decaying into one Higgs boson $H$, which decays in a pair $b\bar{b}$, and an unknown low mass $X$ boson ($\approx O(10^2)$ GeV) with the ATLAS experiment using data from the Run 2 of LHC is discussed here [3]. The minimal assumption for this process is that the $X$ particle decays into fully hadronic final states, allowing to reconstruct it as a single large-radius (R) jet, i.e. the collection of calorimeter energy deposits, also called constituents, resulting from the hadronization of final state quarks. The AD algorithm is used to select events most compatible with the presence of the $X$ particle among the available recorded events, building the so-called anomaly signal region (SR). It's based on the prediction of a fully unsupervised variational autoencoder (AE) structured in a recurrent way, referred to as a variational recurrent neural network (VRNN) [4]: the network in the training phase learns how to reconstruct very well objects it is trained on based on their features, in this case background jets that mainly arise from QCD dijets interactions, so that it commits an error when it checks for hypothetical signal. This can be exploited as an anomaly score (AS), i.e. a discriminant between background (SM) and signal (BSM) that has a higher value the more anomalous the jet is. The VRNN is trained exclusively on data, to account for QCD MC-data disagreements, using as input a sequence of ATLAS-reconstructed four-vector constituents of jets with $p_\mathrm{T} \geq$ 1.2 TeV. The training objective is written as a MSE loss function, used also at prediction level to compute the AS discriminant per jet from its input and output. As seen in Figure 1a, the AS turns out to be not only sensitive to the two-prong decay of the $X$ boson, but also to more exotic signal hypothesis, which appear to have AS $\geq 0.5$. The final fit is executed on the dijet invariant mass $m_\mathrm{JJ}$ of the final state objects, repeated in overlapping bins of the $X$ mass. No significant deviation were observed with respect to the background-only hypothesis in the anomaly SR. Upper limits on the production cross section $\sigma(pp \rightarrow Y \rightarrow XH \rightarrow q\bar{q}b\bar{b})$ were then computed for the background plus signal hypothesis in model-dependent defined SRs. Figure 1b shows the comparison of the limits between the several SRs, highlighting that the anomaly SR provides exclusion as good as the model-dependent SRs when tested on a two-prong channel, while it performs better when tested on more exotic processes thanks to its general approach.

## 3  Graphs and Graph Neural Networks

As widely known, data inputs given to common Neural Network models require some sort of ordered vector structure in the vast majority of cases. Sometimes, however, this format doesn't catch the possible correlations among data. Graphs, i.e. mathematical objects made of nodes (entities) and their relations (edges) that both have associated features, come to the rescue in these cases. ML architectures built specifically to make predictions on graphs data via their relational nature are referred to as Graph Neural Networks (GNNs) [5]Such models learn during training the vector representation (often called embedding) in a latent feature space for each node of the input graphs, done by means of a message passing mechanism. If we imagine a multi-layer GNN architecture, a specific node embedding is updated at each GNN layer by aggregating the features passed between the target node and the nodes from its closest neigh-

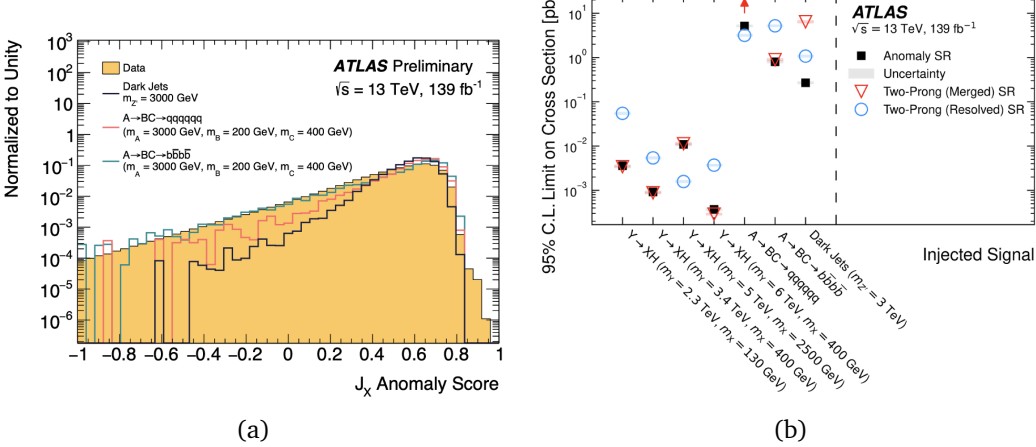

Figure 1: Anomaly score distribution for the $X$ candidate in data and MC signals (a); upper limits on the YXH production cross section at 95% CL, with comparisons between the AD and model-dependent approaches, evaluated over several signal hypotheses. [3]

borhood that is one edge away. Each layer of the GNN added then extends the aggregation of features to a broader neighborhood, in a neighbors of neighbors fashion, effectively increasing the scope of available information. Graph embedding, if needed, can be at trained ultimated computed by pooling each node representation into one global representation, by summing, averaging or finding the max over every vector component. A Transformer [6] is a subclass of GNNs where vector features are updated at each layer mathematically equal to a GNN that takes as input a fully connected graph. Transformers are made up of Attention blocks, which essentially work by skip-summing the input (processed through several linear and normalization layers) and passing it to a scaled dot product attention mechanism. This mechanism splits its input into three branches, with one of them (value) being compared to the result of a combination of the other two (query and key), so that the network pays attention to most important correlations among data during the training phase.

## 4  HEP application: graphs are the new jets

Following the results provided by the YXH search, a new R&D for a fully unsupervised Anomaly Detection approach is being conducted on open data with the prospect of applying it on ATLAS collected data. The main idea consists of representing hadronic jets as graphs, due to their sparse geometry, allowing to exploit their low-level constituents features with GNNs. The search, in fact, looks for a similar process to section 2, defined by a heavy mass resonance Y' decaying into two unknown bosons X and X' which both decay fully hadronically (Figure 2a). Every selected event is assumed to have 2 boosted Large-R jets, due to the kinematic constraints of the explored phase space region. Graph representations of jets are built using as nodes the jets constituents, and using as features the fraction transverse momentum $p_T$ of each constituent and its $\eta$ and $\phi$ coordinates in the ATLAS reference system. Edges, on the other hand, are defined if their distance $\Delta R = \sqrt{\Delta\eta^2 + \Delta\phi^2}$ is less than 0.2, assigning as feature the value $1/\Delta R$. Constituents features also undergo a data augmentation procedure in order to decorrelate the jets mass from a GNN, specifically an Edge-Featured Graph Attention Network (EGAT) model [7], or a Transformer prediction [8], which are the two architectures tested in the R&D phase for now. The Anomaly Detection strategy consists in a Deep Support

Vector Data Description (DeepSVDD) minimization objective [9], in the GNN case, and in a MSE minimization objective in the Transformer case. Both training concepts are data-driven and very similar to the YXH search, where a final AS is assigned to each jet per event based on the distance between data represented in the output space and an input space reference.

## 4.1   Results on LHCOlympics Open Data

The R&D is being carried out on open data from the LHCOlympics 2020 competition [10]. Their dataset consists of a 1.1M events MC simulation of QCD dijets events and a physics process very similar to our searched signal signature, with an assumes 2-prong hadronic decay of the daughter particles and fixed parameters $m_{Y'} = 3.5$ TeV, $m_{X'} = 500$ GeV, $m_X = 100$ GeV. The performance of the AD is evaluated by the AUC of the ROC curve computed from the jet-level AS distribution obtained from the prediction on a validation dataset containing both background and signal graphs, in a 20:1 ratio. Moreover, an event-level AS is also computed by averaging the AS values of both jets found per event. The most promising results, in fact, were given by the EGAT model exploiting this event-level AS, with an AUC around 82% (Figure 2b), also very similar to the performance given by the Transformer architecture, which so far with the jet-level AS reached a 75% of AUC.

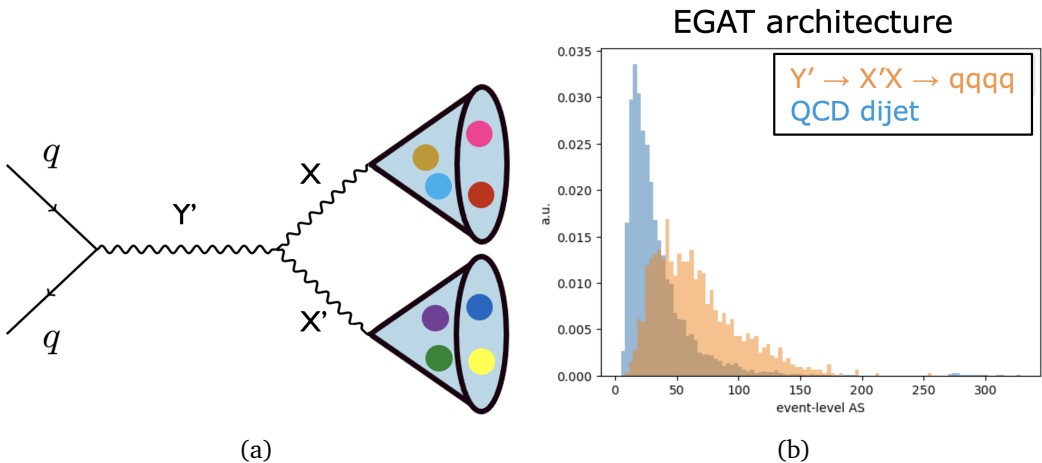

|      (a)      |      (b)      |

Figure 2: Feynman diagram of the BSM physics process from sec. 4 (a); Event-level Anomaly Score distribution, obtained from the output of the EGAT architecture.

## 5   Conclusion

The lack of BSM physics evidences since the Higgs boson discovery [11] contributed to the advancement of more generic searches in the field of High Energy Physics, as opposed to the traditional model-dependent approaches. For this reason, AD as a technique is becoming a mainstream topic at the ATLAS collaboration, particularly in the context of model agnostic searches with jets in the final state that have been conducted and are being conducted by the group. Both works in sections 2 and 4 show promising results. Further progression of the R&D foresees the application of these same architectures on Run 3 ATLAS data along with possible improvements. In particular, the mass and other ATLAS-specific low-level properties of the constituents could be added to the set of node features. Furthermore, an actual event-level graph definition could boost the signal-to-background discrimination, taking advantage of jets correlations by using both jets constituents for the graph building.

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
