# Peer review of "Unsupervised Machine Learning for Anomaly Detection in LHC Collider Searches"

_SciPost Physics Proceedings_

## Round 1 · Referee Report · Anonymous (Referee 1) · 2025-11-19

Strengths

1- promising topic, with significant interest among the experimental collaborations

Weaknesses

1- language needs improving 2- the paper could focus more on the study on LHCOlympics dataset and less on presentation of GNN/transformers (Section 3)

Report

Paper first summarizing an ATLAS published paper a resonance search using anomaly detection techniques. Then GNN and Transformers are tested for anomaly detection on LHCOlympics open dataset. The paper needs language and formatting improvements, and could focus less on presenting ML architectures and more on the work itself (training GNN/Transformers).

Requested changes

1- Section 1, second line : "it's" -> "it is" (throughout) 2- Section 2, first line: "$\approx TeV$" -> consider mentioning it is the mass that is in the TeV range 3- Section 2 : "jet, i.e. the collection of calorimeter energy deposits" : jets in ATLAS are usually made from combination of tracks & calorimeter deposits (particle flow approach). Consider clarifying/dropping that part. 4- Section 3 : missing full stop in 3rd sentence 5- " Graph embedding, if needed, can be at trained ultimated computed by [...]" : consider rephrasing confusing sentence 6- "finding the max" : replace "max" by "maximum" 7- Consider shortening and clarifying Section 3, focusing more on Section 4 (the application) 8- Section 4 : avoid "in fact" 9- Section 4 : Clarify "using as features the fraction transverse momentum". Maybe replace with "fractional transverse momentum" ? 10- Consider explaining the "data augmentation procedure" to decorrelate the jet mass 11- "Constituents features also undergo a data augmentation procedure in order to decorrelate the jets mass from a GNN, specifically an Edge-Featured Graph Attention Network (EGAT) model, or a Transformer prediction, which are the two architecturestested in the R&D phase for now. " : split this sentence into two. 12- "MSE minimization objective in the Transformer case" : consider clarifying what MSE is referring to in unsupervised learning 13- Section 4.1 : "with an assumes" -> "assumed" ? 14- "by the group" : clarify which group is referred to (the ATLAS collaboration ?) 15- references need formatting 16- Figure legend need formatting (subfigures label "(a)" is present but not "(b)")

Recommendation

Ask for minor revision

  • validity: -
  • significance: -
  • originality: -
  • clarity: -
  • formatting: acceptable
  • grammar: below threshold

---

## Editorial Decision

awaiting_resubmission